# Perioperative Cetuximab with Cisplatin and 5-Fluorouracil in Esogastric Adenocarcinoma: A Phase II Study

**DOI:** 10.3390/cancers15072188

**Published:** 2023-04-06

**Authors:** Caroline Gronnier, Christophe Mariette, Come Lepage, Carole Monterymard, Marine Jary, Aurélie Ferru, Mathieu Baconnier, Xavier Adhoute, David Tavan, Hervé Perrier, Véronique Guerin-Meyer, Cédric Lecaille, Nathalie Bonichon-Lamichhane, Didier Pillon, Oana Cojocarasu, Joëlle Egreteau, Xavier Benoit D’journo, Laétitia Dahan, Christophe Locher, Patrick Texereau, Denis Collet, Pierre Michel, Meher Ben Abdelghani, Rosine Guimbaud, Marie Muller, Olivier Bouché, Guillaume Piessen

**Affiliations:** 1Department of Digestive Surgery, Magellan Center, Bordeaux University Hospital, 33600 Pessac, France; denis.collet@chu-bordeaux.fr; 2Department of Digestive and Oncological Surgery, CHU Lille, Claude Huriez University Hospital, 59000 Lille, France; guillaume.piessen@chru-lille.fr (G.P.); 3UMR-S 1172-CANTHER Laboratory “Cancer Heterogeneity, Plasticity and Resistance to Therapies”, University Lille, 59045 Lille, France; 4FFCD EPICAD INSERM LNC-UMR 1231, University of Burgundy and Franche-Comté, 21000 Dijon, France; come.lepage@u-bourgogne.fr; 5Department of Digestive Oncology University Hospital Dijon, University of Burgundy and Franche Comté, 21000 Dijon, France; 6Federation Francophone de Cancérologie Digestive (FFCD), EPICAD INSERM LNC-UMR 1231, University of Burgundy and Franche Comté, 21000 Dijon, France; carole.monterymard@u-bourgogne.fr; 7Department of Digestive Oncology, University Hospital, 63100 Clermont-Ferrand, France; jary.marine@gmail.com; 8Department of Oncology, University Hospital, 86000 Poitiers, France; a.ferru@chu-poitiers.fr; 9Department of Gastroenterology, General Hospital, 74960 Annecy, France; mbaconnier@ch-annecygenevois.fr; 10Department of Gastroenterology, St Joseph General Hospital, 13000 Marseille, France; xadhoute@hopital-saint-joseph.fr; 11Department of Gastroenterology, Lyon Protestant Infirmary Clinic, 69300 Lyon, France; davidtavan@gmail.com; 12Department of Hepato-Gastroenterology, Saint Joseph Hospital, 13000 Marseille, France; hperrier@hopital-saint-joseph.fr; 13Institut de Cancérologie de l’Ouest Paul Papin, 49000 Angers, France; veronique.guerin-meyer@ico.unicancer.fr; 14Department of Hepato-Gastroenterology, Polyclinic Bordeaux Nord, 33000 Bordeaux, France; lecail@hotmail.com; 15Medical Oncology, Clinique Tivoli Ducos, 33000 Bordeaux, France; n.bonichon-lamichhane@tivoli-oncologie.fr; 16Department of Hepato-Gastroenterology, Centre Hospitalier de Bourg en Bresse, 01053 Bourg-en-Bresse, France; dpillon@ch-bourg01.fr; 17Onco-Hematology Department, Centre Hospitalier du Mans, 72000 Le Mans, France; ocojocarasu@ch-lemans.fr; 18Radiotherapy and Oncology Department, Centre Hospitalier Bretagne Sud, 56100 Lorient, France; e.girard@ghbs.bzh; 19Department of Thoracic Surgery, North Hospital, Aix-Marseille University, 13000 Marseille, France; xavier.djourno@ap-hm.fr; 20Service d’Oncologie Digestive, CHU Timone, 13000 Marseille, France; laetitia.dahan@ap-hm.fr; 21Hepato-Gastroenterology Unit, Meaux Hospital, 77100 Meaux, France; clocher@ghef.fr; 22Gastroenterology, Centre Hospitalier de Mont-de-Marsan, 40000 Mont-de-Marsan, France; patrick.texereau@ch-mdm.fr; 23Iron Group, Normandy Centre for Genomic and Personalized Medicine and Department of Hepatogastroenterology, Rouen University Hospital, Normandie University, 76000 Rouen, France; michelphf@gmail.com; 24Department of Oncology, Centre Paul Strauss, 67100 Strasbourg, France; m.ben-abdelghani@icans.eu; 25Centre Hospitalier Universitaire de Toulouse, 31400 Toulouse, France; guimbaud.r@chu-toulouse.fr; 26Department of Gastroenterology, CHU Nancy, 54500 Vandoeuvre-les-Nancy, France; m.muller7@chru-nancy.fr; 27Department of Digestive Oncology, CHU Reims, 51100 Reims, France; obouche@chu-reims.fr

**Keywords:** esogastric adenocarcinoma, neoadjuvant chemotherapy, cetuximab, surgery, efficacy, safety, EGFR-1

## Abstract

**Simple Summary:**

The treatment of resectable gastric and gastroesophageal junction adenocarcinomas is enhanced by a strategy of perioperative chemotherapy (CT) when compared with surgery alone. But, there is still a need for new approaches to further improve outcomes in patients treated with perioperative CT. Cetuximab, a human–murine chimeric monoclonal antibody binds with a high affinity to the EGFR binding site, and has shown activity against a variety of tumors, including G/GEJ adenocarcinomas. This study aimed to evaluate the efficacy and safety of perioperative cetuximab combined with 5-fluorouracil and cisplatin for the treatment of gastric and esophageal adenocarcinoma. The results of this phase two study showed safety but lack of efficacy regarding objective tumor response and absence of major toxicity.

**Abstract:**

Purpose: While perioperative chemotherapy provides a survival benefit over surgery alone in gastric and gastroesophageal junction (G/GEJ) adenocarcinomas, the results need to be improved. This study aimed to evaluate the efficacy and safety of perioperative cetuximab combined with 5-fluorouracil and cisplatin. Patients and Methods: Patients received six cycles of cetuximab, cisplatin, and simplified LV5FU2 before and after surgery. The primary objective was a combined evaluation of the tumor objective response (TOR), assessed by computed tomography, and the absence of major toxicities resulting in discontinuation of neoadjuvant chemotherapy (NCT) (45% and 90%, respectively). Results: From 2011 to 2013, 65 patients were enrolled. From 64 patients evaluable for the primary endpoint, 19 (29.7%) had a morphological TOR and 61 (95.3%) did not stop NCT prematurely due to major toxicity. Sixty patients (92.3%) underwent resection. Sixteen patients (/56 available, 28.5%) had histological responses (Mandard tumor regression grade ≤3). After a median follow-up of 44.5 months, median disease-free and overall survival were 24.4 [95% CI: 16.4–39.4] and 40.3 months [95% CI: 27.5–NA], respectively. Conclusion: Adding cetuximab to the NCT regimen in operable G/GEJ adenocarcinomas is safe, but did not show enough efficacy in the present study to meet the primary endpoint (NCT01360086).

## 1. Background

Although the incidence of gastric cancer adenocarcinoma is declining in the Western world, the number of proximal tumors is increasing, now representing more than 50% of newly diagnosed gastric cancer cases [1,2].

The treatment of resectable gastric and gastroesophageal junction (G/GEJ) adenocarcinomas is enhanced by a strategy of perioperative chemotherapy (CT) when compared with surgery alone [2,3,4], with several chemotherapeutic combinations that have been validated in phase III trials [5,6,7,8]. Following the results of the FFCD9703 trial, a doublet of 5FU and cisplatin has long been considered the standard in France [3,5]. Nevertheless, there is still a need for new approaches to further improve outcomes in patients treated with perioperative CT. To this end, the current challenge is to select novel targeted agents and include them in new treatment strategies for patients with G/GEJ adenocarcinomas [9,10,11]. Several studies have revealed a strong expression of the epidermal growth factor receptor (EGFR), a transmembrane glycoprotein and member of the tyrosine kinase growth factor receptor superfamily, in 32 to 47% of G/GEJ adenocarcinomas [12,13]. Cetuximab, a human–murine chimeric monoclonal antibody (mAb) binds with a high affinity to the EGFR binding site, and has shown activity against a variety of tumors, including G/GEJ adenocarcinomas [13]. Both preclinical and encouraging phase II data suggest a potential benefit of cetuximab, especially in combination with conventional cytostatic therapy in patients with advanced G/GEJ adenocarcinomas [10].

The present trial was set up following the publication of preclinical data and encouraging phase II results suggesting that cetuximab, especially in combination with CT, had a potential benefit in patients with advanced G/GEJ adenocarcinomas [14]. This phase II study was carried out to evaluate the efficacy and toxicity of a regimen combining a targeted therapy, cetuximab, with an established 5-FU-cisplatin perioperative CT regimen for operable G/GEJ adenocarcinomas.

## 2. Patients and Methods

### 2.1. Study Population

Specific inclusion criteria were patients (i) with a histologically proven, previously untreated G/GEJ adenocarcinoma, evaluable according to the response evaluation criteria in solid tumors (RECIST V1.1) [15]; (ii) with a locally advanced tumor (at least stage II or III) on pretreatment examinations but deemed resectable by a multidisciplinary team; (iii) aged 18 to 75 years; and (iv) able to undergo one of the investigated surgical modalities.

Exclusion criteria were (i) malignancy treated within the previous 5 years, (ii) previous abdominal or thoracic radiotherapy, (iii) previous chemotherapy or radiotherapy for G/GEJ adenocarcinoma, (iv) gastric linitis plastica, (v) other concurrent targeted therapy or hormone therapy for cancer, (vi) interstitial pneumonia, and (vii) weight loss exceeding 15% in the 6 months preceding cancer diagnosis.

### 2.2. Study Design and Treatment (Figure 1)

This open-label non-randomized single arm multicenter phase II study was conducted in 25 centers in France; approved by the independent ethics committees of the participating sites; and designed and conducted according to good clinical practice, the Declaration of Helsinki, and all local requirements. All patients gave written informed consent. This study was registered at https://clinicaltrials.gov/, number NCT 01360086.

This study was open to patients with resectable G/GEJ adenocarcinoma. The pre-inclusion workup, neoadjuvant and adjuvant treatment, and clinical response work-up are detailed in Figure 1. Non-traversable tumors were considered cT3N+ [16]. GEJ tumors were classified according to the Siewert classification [17], the clinical tumor–node–metastasis (cTNM) classification was based on usTNM classifications [18].

Surgery was planned to take place 3–4 weeks after the end of neoadjuvant CT. Details of the surgical resection with an aim of R0 resection have been described elsewhere [19,20]. Histology was analyzed according to the 7th pTNM classification [21]. The tumor regression grade was evaluated according to the Mandard classification [22]. Patients were followed for 1 month after surgery. Post-operative morbidity was assessed according to the 2004 Dindo–Clavien classification. Complications ≥ grade II were considered significant complications [23].

Patients were followed after the end of adjuvant CT, then every 4 months during the first 2 years and every 6 months during the third year with clinical examinations and thoracoabdominal CT-scans. Health-related QoL was prospectively assessed using QLQ-C30 version 3.0 [24] and STO22 [25] before inclusion, one month after surgery, and then every 4 months until the end of treatment.

**Figure 1 cancers-15-02188-f001:**
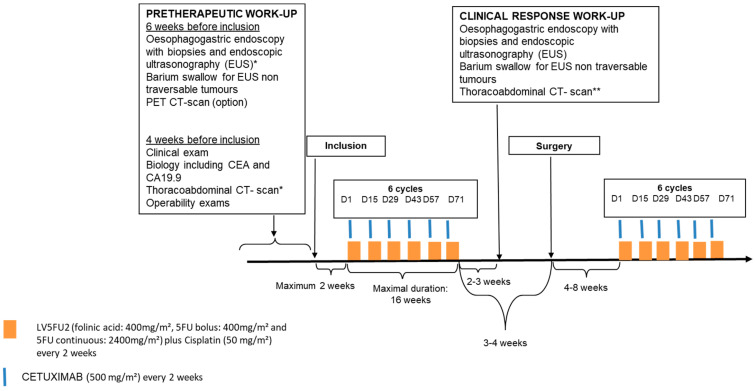
Therapeutic strategy. GEJ tumours were classified according to the Siewert classification [15].* The clinical tumor–node–metastasis (cTNM) classification we used was based on usTNM classifications for carcinoma of the esophagus [19] and the stomach [20]. Non-traversable tumors were considered cT3N+ as previously published [16].** CT-scans were centrally reviewed according to the RECIST V1.1 criteria by an independent observer blinded to the outcomes prospectively in one center.

### 2.3. Evaluation Criteria

The combined primary end point consisted of the tumor objective response (TOR) according to RECIST V1.1 criteria on CT scans (patients with a complete or partial response) [15] and major toxicities resulting in discontinuation in the neoadjuvant CT setting. CT treatment was stopped in the following cases: investigator’s decision, or major toxicity or adverse event (AE), or disease progression, or patient’s decision.

The secondary endpoints were the R0 resection rate, treatment tolerability, postoperative morbi-mortality, 1- and 2-year recurrence rate, 3-year recurrence-free survival (RFS), disease-free survival (DFS), overall survival (OS), QoL, and correlations between the response rate and skin toxicity grade.

The safety and tolerability of cetuximab in combination with 5FU and cisplatin were assessed by recording AE during the treatment period. AE were graded according to the National Cancer Institute Common Toxicity Criteria Version 4.0 (NCI-CTC v4.0) [26]. RFS, DFS, and OS were defined as the time from inclusion to first disease recurrence or death, first recurrence or second cancer or death, and all-cause death, respectively. Health-related QoL was assessed using QLQ-C30 version 3.0 [24] and STO22 [25]. A five-point difference was considered the smallest clinically significant difference. The higher the score for global health status and physical functioning, the better the QoL, and the higher the score for symptoms, the worse the QoL. The time until the first deterioration in QoL was the time between inclusion and the date of a decrease of more than five points in the global health score. All of these data were reviewed by an independent multidisciplinary committee.

### 2.4. Statistical Analysis

Efficacy criteria were TOR and major toxicities resulting in discontinuation of the neoadjuvant CT, evaluated before surgery. The sample size was calculated using Bryant and Day design in two steps (α = 5%, power = 80%), N = 65, expecting a rate of TOR of 45% and a percentage of patients without major toxicities of 90%.

The decision rules were as follows: if ≤22 patients showed an objective response, the treatment would be considered of no interest; if ≤52 patients showed no major toxicity, the treatment would be considered of no interest (excessive toxicity). If more than 22 enrolled patients showed a TOR, an interim analysis would be done leading to continuation of the study.

All of the analyses were conducted in an intent-to-treat (ITT) population. We assessed tolerance and safety in all patients who received at least one dose of the study drug (safety population). All data were reported using usual descriptive statistics: for quantitative variables: means, standard deviations, medians, inter-quartile intervals, and ranges (minimum and maximum) and for qualitative variables, frequencies and percentages. The results were also described with WaterFall plots, representing for each patient the percentage tumor size change since the first evaluation during inclusion. This percentage was represented on the horizontal axis according to RECIST criteria. Survival analyses were done using the Kaplan–Meier method and described using medians with two-sided 95% confidence intervals (CI). DFS was estimated with a landmark method. An exploratory analysis was carried out to investigate the impact of skin rash on tumor response. Analyses were done using SAS 9.4 (SAS Institute, Cary, NC, USA).

## 3. Results

### 3.1. Patients

The flow chart of the trial is shown in Figure 2. From June 2011 to March 2013, 65 patients were recruited. Sixty-four patients were evaluable for primary endpoint analysis, and one patient had no evaluable preoperative workup. The patient characteristics are shown in Table 1. Most patients were male (83.1%); with a WHO performance status of 0 (63.1%); and had stage III G/GEJ adenocarcinoma (71.4%), located mainly at the GEJ (69.2%).

### 3.2. Neoadjuvant Chemotherapy

All patients started CT and received at least one cycle of neoadjuvant CT. The median number of cycles per patient was 6.0 [range: 1–6]. Fifty-eight patients (89.2%) completed all cycles of neoadjuvant CT. The cumulative doses and dose modifications are shown in Appendix A. Forty-six patients (70.8%) received a combination of the three drugs at each cycle. Dose reduction was necessary for cetuximab in 14 patients (21.5%), cisplatin in 22 (33.8%), and 5FU in 23 (35.4%). Dose delays occurred in 36 (55.4%) patients. Twenty-nine (44.6%) had one delay, six (9.2%) had two delays, and one (1.5%) had three delays.

The toxicities of neoadjuvant CT are detailed in Table 2. Grade 3 or 4 AEs were observed in 40 patients (61.5%). Only neutropenia was a grade 4 toxicity occurring in seven patients (10.8%). No deaths due to toxicity were observed. Acneiform rash was observed in 92.3% of patients, but was grade 3–4 in only 7.7% of patients. Paronychia, with a maximal grade 1–2, was observed in 3.1% of cases. A reaction/hypersensitivity, with a maximal grade 1–2, and hypomagnesemia were observed in 1.5% of cases. The median time to the onset of ≥grade 3 toxicity was 70 days [95% CI: 42–115]. Seven patients stopped the treatment prematurely: three for major toxicity, two for events linked to the tumor (dysphagia and tumor ulceration), and two on the investigator‘s decision (treatment not administered). The three major toxicities were attributed by the investigator to (i) thrombopenia due to cisplatin, (ii) neutropenia and mucositis due to 5FU and cisplatin, and (iii) neutropenia due to 5FU and cetuximab in one case.

### 3.3. Primary Endpoint: Major Toxicities Leading to Discontinuation of Neoadjuvant CT and TOR

Neoadjuvant CT was safe, as 61 patients (95.3%) (≥52 patients) showed no major toxicity resulting in definitive discontinuation of the neoadjuvant CT. TOR was evaluable in 64 patients and is detailed in Figure 3. Responses analyzed centrally were the same as those analyzed at the center in 35 cases (54.7%) (detailed in Appendix A). No patient had a complete response. Stable disease (SD) was recorded in 37 (57.8%) patients and progressive disease in eight patients (12.5%). Nineteen (29.7%) of the 64 evaluable patients had an objective morphological tumor response. According to the decision rules defined in the protocol, neoadjuvant treatment was considered of no interest, as only 19 patients showed an objective response (≤22 patients). There was no statistical relationship between the objective response grade and AE due to skin rash during neoadjuvant CT, evaluated through logistic regression (*p*-value = 0.5340).

### 3.4. Surgical Treatment and Postoperative Outcomes

Sixty patients (92.3%) underwent surgical resection (Figure 2). The surgical and pathological results for the 60 resected patients are shown in Table 3. Six patients had a poorly cohesive carcinoma and seven had a major component of poorly cohesive carcinoma. Given the tumor location, most patients underwent an Ivor Lewis procedure (63.4%). Twenty-five (41.7%) patients had significant postoperative complications and two (3.3%) postoperative deaths occurred. Resection was R0 in 89.8% of cases. Most patients had pT3 disease (63.3%) with lymph node involvement (61.7%). Upon histological analysis, 16 patients (out of 56 available) (28.6%) were responders (Mandard tumor regression grade ≤ 3).

### 3.5. Adjuvant Chemotherapy

Forty-nine patients (75.4%) started adjuvant CT. Of all of the patients included, 31(47.7%) patients received the complete scheduled treatment. One patient received adjuvant oxaliplatin instead of cisplatin due to weakness after surgery. The cumulative doses and dose modifications are shown in Appendix A. Dose reduction was necessary for cetuximab in 17 cases (35.4%), cisplatin in 29 (60.4%), and 5FU in 27 (56.3%). Toxicity data are detailed in Table 2. Grade 3/4 AEs were observed in 25 patients (51.0%). Among the 15 premature discontinuations of adjuvant CT, 10 patients stopped for major toxicity (66.7%), four for disease progression (26.7%), and one for an inter-current event (6.7%). The reasons for treatment discontinuation for major toxicity were: poor general condition (n = 6), renal insufficiency (n = 2), and hematologic toxicity (n = 2). The median time to the onset of grade 3/4 toxicity was 111 days [50; not reached].

### 3.6. Efficacy

The median follow-up was 44.5 months [95% CI—43.1; 56.2]. The 1- and 2-year recurrence rate was 5.5% (3 patients) and 23.6% (13 patients), respectively. Median time to recurrence was 22.1 months [95% CI—16.3; NR]. The three-year recurrence-free survival was 54.8% [95% CI—41.1; 66.2] with a median of 42.7 months [95% CI—23.1; NR]. Three-year DFS was 39.9% [95% CI—28.0; 51.5] with a median of 24.4 months [95% CI—16.4; 39.4]. Three-year OS was 55.1% [95% CI—42.2; 66.3] with a median of 40.3 months [95% CI—27.47; NR].

### 3.7. Health-Related Quality of Life

Health-related quality of life is detailed in Appendix A. Sixty-three patients (96.9%) completed at least one questionnaire; the median number of questionnaires answered per patient was six [range: 1−12]. The global health status deteriorated in 38 patients (63.3%) after a median time of 7.9 months [95% CI—3.8; 19.2]. Twenty-three (38.3%) patients showed a deterioration in the fatigue score, with a median time not reached. The median survival with no deterioration in the global health status was 5.6 months [95%CI—3.8; 9.8]. Rates of survival with no deterioration in the global health status at 1, 2, 4, 6, 12, and 24 months were 98.3% [95%CI—88.8; 99.8], 96.7% [95%CI—87.3; 99.2], 61.7% [95%CI—85.3; 98.4], 46.7% [95%CI—33.7; 58.6], 31.7% [95%CI—20.4; 43.5], and 20.0% [95%CI—11.0; 30.9], respectively.

## 4. Discussion

In this phase II study, the addition of cetuximab to the neoadjuvant CT regimen including 5FU-cisplatin in operable G/GEJ adenocarcinomas was safe but did not show enough efficacy to meet the primary endpoint, which was a combined evaluation of tumor response according to RECIST criteria and major toxicities resulting in discontinued neoadjuvant CT. The computed tomography criteria were assessed centrally. Only 29.7% of patients had an objective morphological tumor response (based on the centralized review), which is comparable to previous reports of neoadjuvant regimens with 5FU and cisplatin without cetuximab [5,27], and 95.3% did not stop their treatment prematurely because of major toxicity.

At the time this study was designed, several phases II studies focusing on metastatic and advanced forms of gastric cancer had suggested a potential benefit of adding cetuximab to CT [10,14,28,29], with an overall response rate (ORR) of 41.2% with cisplatin and docetaxel [14], between 44.1% [28] and 46% [10] with 5FU and irinotecan (FOLFIRI), and 65% with oxaliplatin-based CT 65% [29]. Randomized phase III trials evaluating the interest of adding EGFR-targeted agents (cetuximab in the EXPAND trial [30] and panitumumab in the REAL3 [31] and MEGA [32] trials) in combination with CT failed to demonstrate any benefits in patients with metastatic and advanced gastric cancer. The REAL-3 trial even reported significantly inferior survival following the addition of an EGFR agent [31].

Other studies have focused on the addition of cetuximab to neoadjuvant radiochemotherapy for locally advanced esophageal cancer. A phase I/II trial evaluating two cycles of induction cetuximab with cisplatin and docetaxel followed by biochemoradiation (45 Gy plus concurrent cisplatin and cetuximab) and a phase II trial evaluating two weekly doses of cetuximab in monotherapy followed by weekly cetuximab combined with radiation therapy showed favorable results with a limited toxicity [33,34]. Conversely, the combination of cetuximab with irinotecan, cisplatin, and radiotherapy was considered toxic, with severe neutropenia occurring in 47%. Moreover, the pCR rate with the combination was insufficient to merit further evaluation [35]. The results of the SAKK phase III trial, in which the anti-EGFR agent cetuximab was added to preoperative chemoradiotherapy, showed improved local tumor control with reduced local tumor recurrence. However, the differences in progression-free and overall survival were not significant. The reduction in local tumor recurrence occurred despite no differences in either a successful R0 resection or a pathologic complete response at surgery [33]. The RTOG 04362 and SCOPE-1 trials looked at cetuximab combined with non-operative chemoradiotherapy in patients with esophageal cancers. It showed no advantage in overall survival or tumor response, thus demonstrating no putative beneficial effect of cetuximab on radiation therapy [36,37].

Our study is the first to evaluate the interest regarding adding cetuximab to a perioperative chemotherapy strategy. The choice of including gastric, esogastric junction, and esophageal cancers was supported by previously published phase III studies [5,6], and has been confirmed as a valid option in the recently published FLOT 4 trial [8]. In this study, preoperative grade 3/4 cancers were present in 40 patients (61.5%), and the most prominent grade 3/4 treatment-related adverse effects were neutropenia in 18 (27.7%) patients, anorexia in 6 (9.2%), and skin toxicity in 6 (9.2%). This rate of grade 3/4 AE was higher than in the ACCORD-07 trial (38% grade 3–4), but this may also be explained by the regimen of 5FU and cisplatin, which were different in the two studies [5]. Unlike the findings in metastatic colorectal cancer, we found no correlation between skin toxicity and tumor response to cetuximab [38].

Our primary endpoint combined tumor response and toxicity. As we only had data from metastatic studies evaluating TOR, we could only generate our hypothesis based on this finding. One major strength of this study was the centralized reassessment of data, which led to a modification in nearly half of the cases (46.2%). Toxicity is a key point when considering a perioperative strategy. However, histological response, a more objective evaluation of tumor response, was analyzed in this study and led to a pCR response of 7.1%. When compared with trials reporting results of combinations of 5FU and cisplatin without cetuximab, our results were similar to those of the EORTC-40954 trial [27], but slightly higher than in the ACCORD-07 (3%) and MRC-05 (1%) trials [5,7]. Alderson et al. reported comparable surgical approaches in the OE05 trial with a vast majority of transthoracic esophagectomies and similar post-operative outcomes in the 5FU-cisplatin arm, suggesting that the addition of cetuximab did not worsen postoperative outcomes [7]. The 3-year results observed in the trial were better than in the OE-05 trial, but similar to the 5FU-cisplatin arm of the PRODIGE-07 and EORTC-40954 trials [5,27].

Moreover, the level of EGFR amplification and overexpression predicted the response and survival benefit in a preclinical gastric cancer trial treating patient-derived xenografts with cetuximab [39]. In a model-based analysis of response and resistance factors of cetuximab treatment in gastric cancer cell lines, the model predicted the effect of MET mutations on cetuximab sensitivity [40].

Given the consistently negative results for EGFR-targeted therapy in esophago−gastric cancer, except for a biomarker-driven clinical trial, it seems unjustifiable to promote these agents. New data from the Cancer Genome Atlas research network show that the largest subgroup of gastric tumors is the group of chromosomally unstable tumors, which account for half of all cases (65% of junctional tumors) and correlate well with a pathological intestinal phenotype. Within this group, several targetable pathways have been identified, including the epidermal growth factor receptor (EGFR) [41]. Intriguing retrospective biomarker analyses from the COG trial suggest that a subpopulation of tumors with EGFR copy number gain may be susceptible to anti-EGFR therapy, implying that refining the EGFR biomarker may yet yield positive results [42]. As this study was set up, 5FU cisplatin is no longer the most widely used regimen for the treatment of G/GEJ adenocarcinomas, and FLOT has become the gold standard [8]. Further studies should include this regimen.

## 5. Conclusions

The FFCD 0901 trial evaluated the addition of cetuximab to neoadjuvant CT, and this strategy did not lead to premature withdrawal from the study. However, the regimen offered no benefit in terms of efficacy. Based on the results of this phase II study, the use of perioperative cetuximab in combination with 5FU-cisplatin cannot be recommended in an unselected population with resectable G/GEJ adenocarcinomas.

## Figures and Tables

**Figure 2 cancers-15-02188-f002:**
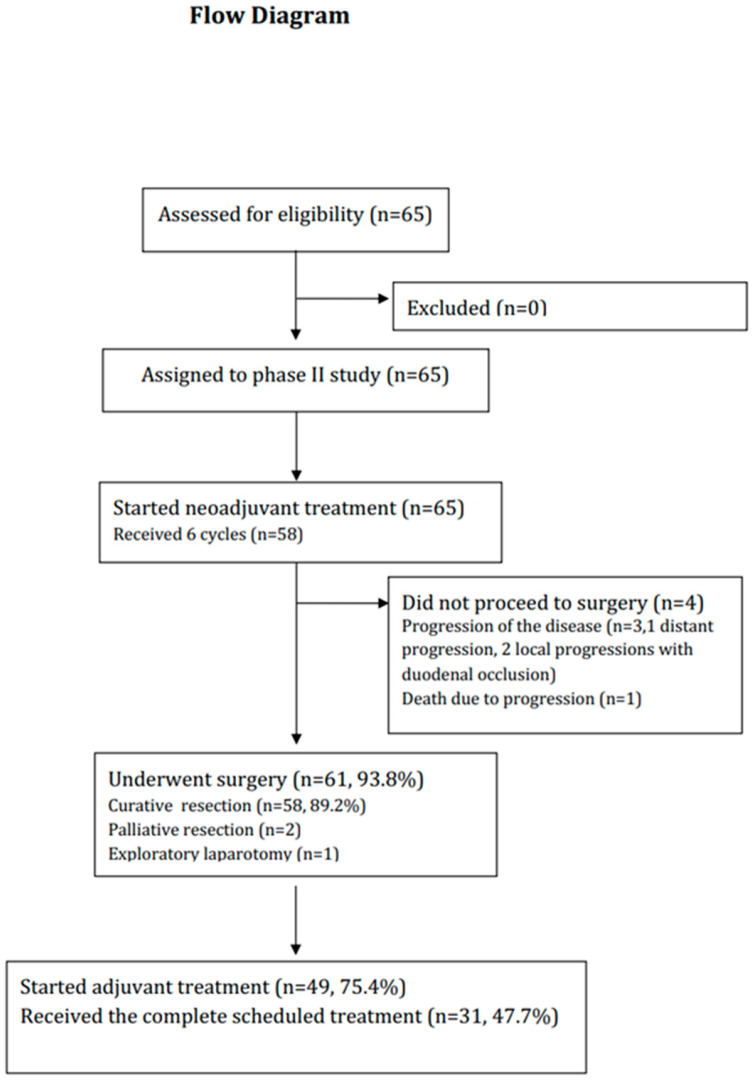
Flow chart of the study.

**Figure 3 cancers-15-02188-f003:**
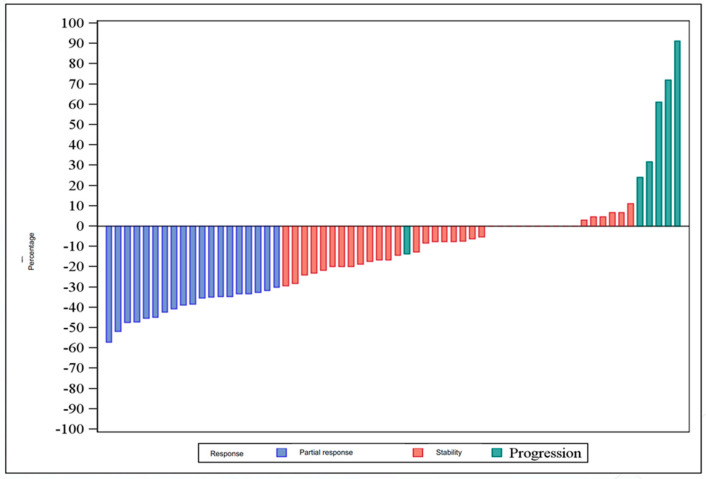
Waterfall graph: percentage change in tumor size according to RECIST criteria.

**Table 1 cancers-15-02188-t001:** Patient and tumor characteristics at baseline.

Characteristics	Patients, N = 65 (%)
Sex	
Male	54 (83.1)
Median age, years (range)	60.5 (40.6–76.7)
WHO Performance Status	
0	41 (63.1)
1	23 (35.4)
2	1 (1.5)
Malnutrition	11 (16.9)
Primary tumor location	
Junctional cancer	45 (69.2)
Siewert type I	22 (48.9)
Siewert type II	20 (44.4)
Siewert type III	3 (6.7)
Gastric cancer	20 (30.8)
Fundus	3 (15.0)
Body	6 (30.0)
Pyloric antrum	11 (55.0)
Echoendoscopy done	62 (95.4)
traversable tumor	49 (79.0)
usT stage	
usT1	1/49 (2.0)
usT2	7/49 (14.3)
usT3	40/49 (81.6)
usT4	1/49 (2.0)
usN stage	
usN0	8/49 (16.3)
usN1	41/49 (83.7)
Clinical tumor stage at inclusion	
II	14/49 (28.6)
III	35/49 (71.4)
Poorly cohesive cells on biopsies	l0/61 (16.4)

WHO: World Health Organization; Malnutrition: Weight loss > 10% in 3 months, Siewert classification [15].

**Table 2 cancers-15-02188-t002:** Potential chemotherapy-associated adverse events during neoadjuvant and adjuvant treatment.

	Neoadjuvant Treatment		Adjuvant Treatment	
AE	Grade 1–2 N = 65 (%)	Grade 3–4 N = 65 (%)	Grade 1–2 N = 49 (%)	Grade 3–4 N = 49 (%)
Diarrhea	21 (32.3)	3 (4.6)	26 (53.1)	1 (2.0)
Anorexia	35 (53.8)	6 (9.2)	26 (53.1)	9 (18.4)
Vomiting	28 (43.1)	4 (6.2)	20 (40.8)	4 (8.2)
Nausea	42 (64.6)	4 (6.2)	32 (65.3)	3 (6.1)
Dysphagia	7 (10.8)	4 (6.2)	1 (2.0)	-
Constipation	28 (43.1)	2 (3.1)	3 (6.1)	-
Stomatitis or mucositis	36 (55.4)	5 (7.7)	18 (36.7)	1 (2.0)
Dyspnea	4 (6.2)	1 (1.5)	4 (8.2)	2 (4.1)
Asthenia	49 (75.4)	6 (9.2)	35 (71.4)	8 (16.3)
Acneiform rash	55 (84.6)	5 (7.7)	33 (67.3)	4 (8.2)
Hand-foot syndrome	6 (9.2)	1 (1.5)	2 (4.1)	-
Xerosis	23 (35.4)	-	11 (22.4)	-
Leukopenia	37 (56.9)	4 (6.2)	25 (51.0)	1 (2.0)
Neutropenia	31 (47.7)	18 (27.7)	26 (53.1)	5 (10.2)
Anemia	53 (81.5)	1 (1.5)	45 (91.8)	1 (2.0)
Thrombocytopenia	35 (53.8)	-	16 (32.7)	-
Serum AST	5 (7.7)	-	3 (6.1)	-
Serum ALT	3 (4.6)	-	7 (14.3)	-
Serum GGT	17 (26.2)	1 (1.5)	18 (36.7)	1 (2.0)
Alkaline phosphatase	6 (9.2)	-	8 (16.3)	-
Fever	8 (12.3)	-	2 (4.1)	-
Peripheral neuropathy	7 (10.8)	-	4 (8.2)	1 (2.0)
Pain	16 (24.6)	-	10 (20.4)	1 (2.0)
Alopecia	14 (21.5)	-	8 (16.3)	-
Renal insufficiency	1 (1.5)	1 (1.5)	3 (6.1)	2 (4.1)
Infection	4 (6.2)	1 (1.5)	4 (8.2)	2 (2.1)
Thrombosis	3 (4.6)	3 (4.6)	-	1 (2.0)
Toxic death	-	-	-	-

ITT population (n = 65), none had grade 5 toxicities; AE, Adverse Events; ITT, intent-to-treat; AST: aspartate aminotransferase; ALT: alanine aminotransferase.

**Table 3 cancers-15-02188-t003:** Surgical and pathology results in the resected population (n = 60).

	N (%), n = 60
Surgical procedure	
Ivor Lewis procedure	38 (63.4)
Total esogastrectomy with intrathoracic anastomosis	13 (21.7)
Distal gastrectomy	5 (8.3)
Total gastrectomy	4 (6.7)
30-day overall morbidity (n = 59)	33 (55.9)
Significant postoperative complications	25 (41.7)
30-day postoperative deaths	2 (3.3)
Surgical complications	16 (26.7)
-anastomotic leak	4
-vocal cord paralysis	2
-intra-abdominal abscess	3
-wound infection	2
-gastroplasty necrosis	2
-small bowel occlusion	2
-chylothorax	2
-pancreatitis	1
-repeat surgery	8
Medical complications (%, n = 59)	14 (23.7)
-pulmonary	11 (18.6)
-cardiovascular	6 (10.2)
-neurologic	3 (5.1)
-renal	2 (3.4)
-hepatic	0 (0)
-sepsis	4 (6.8)
Tumor differentiation (%, n = 55)	
Good	15 (27.3)
Moderate	18 (32.7)
Poor	22 (40.0)
Median number of harvested lymph nodes	25 [7–60]
Median percentage of invaded lymph nodes among harvested lymph nodes	4.9 % [0–73]
Radicality of resection (%, n = 59)	
R0	53 (89.8)
R1	5(8.5)
R2	1 (1.7)
Tumor response (%, n = 56)	
pCR: no residual tumor cells	4 (7.1)
pPR: rare residual tumor cells or more fibrosis without tumor cells	12 (21.4)
pNR: more tumor cells than fibrosis or no histologic sign of response to chemotherapy	40 (71.4)
pT0	3 (5.0)
pT1	7 (11.7)
pT2	5 (8.3)
pT3	38 (63.3)
pT4a	6 (10.0)
pT4b	1 (1.7)
pN0	23 (38.3)
pN1	14 (23.3)
pN2	11 (18.3)
pN3a	11 (18.3)
pN3b	1 (1.7)
pM0	59 (98.3)
pM1	1 (1.7)

## Data Availability

Data available upon request. All patients gave written informed consent. Study design approved by an independent ethics committee.

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
