# Peer review of "Perioperative Cetuximab with Cisplatin and 5-Fluorouracil in Esogastric Adenocarcinoma: A Phase II Study"

_cancers, 2023, doi:10.3390/cancers15072188_

Round 1

Reviewer 1 Report

The paper was well written. The data were fine to support the hypothesis. The language should be improved. The logic should be adjusted in the background section. The conclusion section is required to be heavily revised, the current one is informal.

Author Response

We thank you very much for the time you spent in proofreading and correcting our article and for the valuable comments.

The paper was well written. The data were fine to support the hypothesis.

The language should be improved : 

We thank the reviewer for his comment and suggestion. Thus, we had the article proofread by a professional English language proofreader Mr. Bastable, and the corrections were included in the text.

The logic should be adjusted in the background section. The conclusion section is required to be heavily revised, the current one is informal.

We thank the reviewer and have made changes in the text.

Reviewer 2 Report

This study focused on perioperative cetuximab-containing treatments for GC or EGJ. This article has a poor background for the development of this regimen scientifically. Additionally, these results have no new findings for experts on treatments for GC.

Author Response

We thank you very much for the time you spent in proofreading and correcting our article and for the valuable comments.

This study focused on perioperative cetuximab-containing treatments for GC or EGJ. This article has a poor background for the development of this regimen scientifically. : 

 We thank the reviewer for his comment and have added the following justification to the text on page 5:

Both preclinical and encouraging phase II data suggest a potential benefit of cetuximab especially in combination with conventional cytostatic therapy in patients with advanced G/GEJ adenocarcinomas . 10

Additionally, these results have no new findings for experts on treatments for GC: 

Indeed, this negative phase II trial did not lead to a phase 3 trial. However, we believe it is essential that the results of the negative trials be published so that experts in gastric cancer treatment can explore other therapeutic and research options.

Reviewer 3 Report

Perioperative cetuximab with cisplatin and 5-fluorouracil in esophagogastric carcinoma: a phase II study.

Major:

The authors provide interesting work on adding Cetuximab to a perioperative chemotherapy in patients with locally advanced AEG. Although there exist exhausting negative data about cetuximab therapy in several other settings in AEG, no data have been presented in the present perioperative setting. 

My deepest concern consists 

1)    Age of data, as the study was performed between 2011 and 2013;

2)    No molecular data are presented that may highlight and explain the results.

Thus, I recommend to discuss the relevance of parameters/ mutations that may influence effectiveness of therapy wit cetuximab.

Language is profound except of some smaller inconsistencies (eg page 7: “.. . grade ¾.. “)

Please, get the data / manuscript checked by a statistician as I am a clinician.

Author Response

We thank you very much for the time you spent in proofreading and correcting our article and for the valuable comments.

Major:

The authors provide interesting work on adding Cetuximab to a perioperative chemotherapy in patients with locally advanced AEG. Although there exist exhausting negative data about cetuximab therapy in several other settings in AEG, no data have been presented in the present perioperative setting. 

My deepest concern consists 

1)    Age of data, as the study was performed between 2011 and 2013;

2)    No molecular data are presented that may highlight and explain the results:

Thus, I recommend to discuss the relevance of parameters/ mutations that may influence effectiveness of therapy wit cetuximab.

We thank the reviewer for his comment, a molecular ancillary analysis had been planned but unfortunately could not be completed due to organizational and personal reasons (death of principal investigator Pr Christophe Mariette).

We have as suggested added a paragraph on the subject in the discussion part.

The level of EGFR amplification and overexpression predicted response and survival benefit in a preclinical gastric cancer trial treating patient‑derived xenografts with cetuximab (Wang Oncol Rep 2017). In a Model-based analysis of response and resistance factors of cetuximab treatment in gastric cancer cell lines, the model predicted the effect of MET mutations on cetuximab sensitivity.(Raimundez PLoS Comput Biol 2021).

Language is profound except of some smaller inconsistencies (eg page 7: “.. . grade ¾.. “)

This has been fixed.

Please, get the data / manuscript checked by a statistician as I am a clinician : 

We thank the reviewer, this has been achieved. Indeed, Dr. Carole Monterymard, co-author of the article, is a statistician and has entirely reviewed the data.

Round 2

Reviewer 2 Report

This paper focuses on EGFR antibody plus chemotherapy for resectable EC.

Nowadays, there are little interesting for EGFR antibody for experts, therefore, this paper dose not have interesting items.